# UniGO: A Unified Graph Neural Network for Modeling Opinion Dynamics on Graphs

## ABSTRACT

Polarization and fragmentation in social media amplify user biases, making it increasingly important to understand the evolution of opinions. Opinion dynamics provide interpretability for studying opinion evolution, yet incorporating these insights into predictive models remains challenging. This challenge arises due to the inherent complexity of social interactions, the diversity of opinion fusion rules, and the difficulty in capturing equilibrium states while avoiding over-smoothing. This paper introduces UniGO, a unified framework for modeling opinion evolution on graphs. By abstracting various opinion dynamics models into a unified graph-based structure, UniGO captures both common features and complex fusion rules. Using a coarsen-refine mechanism, UniGO efficiently models opinion dynamics through a graph neural network, mitigating over-smoothing while preserving equilibrium phenomena. Additionally, UniGO leverages pretraining on synthetic datasets, which enhances its ability to generalize to real-world scenarios, providing a viable paradigm for large-scale applications of opinion dynamics. Experimental results on both synthetic and real-world datasets demonstrate UniGO's effectiveness in capturing complex opinion formation processes and predicting future evolution. The pretrained model also shows strong generalization capability, validating the benefits of using synthetic data to boost real-world performance.

## CCS CONCEPTS

• **Mathematics of computing → Ordinary differential equations**; • **Computing methodologies → Neural networks**; • **Applied computing** → *Sociology*.

## KEYWORDS

Opinion Dynamics, Graph Nerual Networks, Graph Coarse, Social Networks

**ACM Reference Format:**
Anonymous Author(s). 2018. UniGO: A Unified Graph Neural Network for Modeling Opinion Dynamics on Graphs. In *Proceedings of Make sure to enter the correct conference title from your rights confirmation emai (Conference acronym 'XX)*. ACM, New York, NY, USA, 10 pages. https://doi.org/XXXXXXX.XXXXXXX

## 1 INTRODUCTION

Polarization in online social media fosters user bias and hostility, exacerbating societal divisions and undermining social cohesion. In recent years, the formation of opinions on social media platforms has emerged as an increasingly important issue. Opinion dynamics quantify individual views and model group opinion evolution during interactions using methods such as agent-based models and ordinary differential equations [18]. As shown in fig. 1a, in large-scale online social media scenarios, opinion interactions are constrained by an underlying graph topology. Opinion dynamics over graph structures have been extensively applied in such contexts, playing a significant role in social network analysis [7, 30], marketing [29, 40], recommendation systems [19], and various other domains.

In contrast to data-driven opinion prediction models, opinion dynamics models focus on interpretability, providing mechanisms for opinion formation. Opinion dynamics explore the mechanisms of group opinion evolution by designing specific opinion fusion rules [2, 8, 22], such as the Friedkin-Johnsen (FJ) model, which considers individuals' stubbornness towards their own opinions [21], and the Hegselmann-Krause (HK) model, which is based on a confidence threshold for others' opinions [24]. Additionally, opinion dynamics investigate equilibrium phenomena within the system, such as consensus, polarization, and fragmentation, implying that opinions cease to change over time [5]. Opinion dynamics offer deep insights into the process of opinion formation. However, integrating these insights into data-driven prediction models remains an open challenge.

Inspired by works such as physics-guided neural networks and graph ODEs [26, 39, 44], recent research has explored the integration of neural networks with opinion dynamics [36]. Graph neural networks (GNNs) have emerged as a natural choice for learning opinion dynamics on graphs. As shown in fig. 1b, there are notable similarities between the opinion update process in single-step dynamics and node embedding updates in single-layer GNNs. However, integrating insights from opinion dynamics into GNNs faces two primary challenges: (1) **Learning diverse opinion fusion rules.** Unlike dynamical systems in the physical world, opinion evolution data in social media is sparse and limited in availability. Using synthetic data generated from theoretical models to enhance model generalization has become an important research direction for data-driven methods in recent years. Different models design different opinion fusion rules based on distinct assumptions, allowing them to study the evolution of group opinions under specific rules. Integrating each of these assumptions into a single model is considerably limited. (2) **Capturing specific equilibrium phenomena.** Through multiple rounds of opinion updates, opinion dynamics effectively describe equilibrium phenomena during opinion evolution, while GNNs may encounter over-smoothing with a high number of layers [35]. Learning the equilibrium of opinions while avoiding over-smoothing remains a significant challenge.

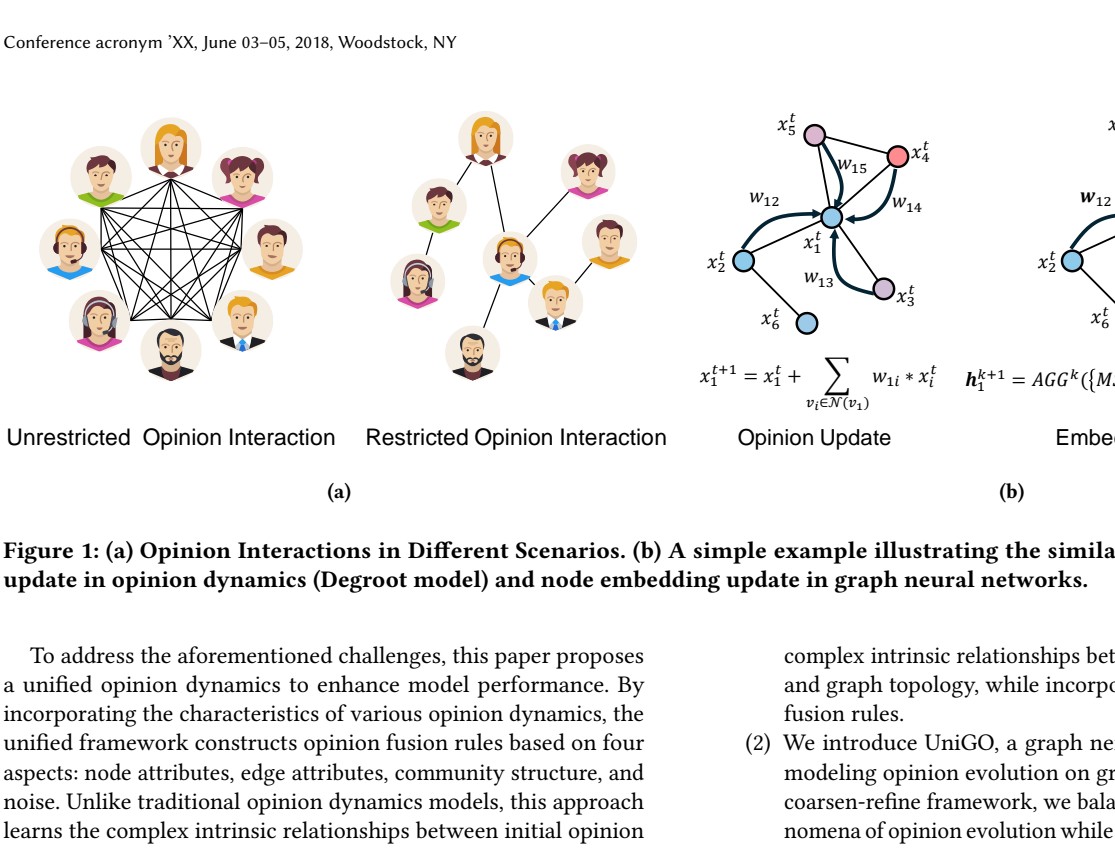

Figure 1: (a) Opinion Interactions in Different Scenarios. (b) A simple example illustrating the similarity between opinion update in opinion dynamics (Degroot model) and node embedding update in graph neural networks.

To address the aforementioned challenges, this paper proposes a unified opinion dynamics to enhance model performance. By incorporating the characteristics of various opinion dynamics, the unified framework constructs opinion fusion rules based on four aspects: node attributes, edge attributes, community structure, and noise. Unlike traditional opinion dynamics models, this approach learns the complex intrinsic relationships between initial opinion formation and graph topology, enabling better predictions of future opinion evolution. Although the unified model does not directly analyze the impact of individual properties on opinion evolution, it facilitates the learning of complex opinion fusion rules by neural networks.

In addition, this paper addresses the second challenge using a coarsen-refine mechanism that applies graph neural networks on the skeleton of the underlying graph. We utilize graph pooling to construct a skeleton structure from the original nodes to supernodes, and use graph neural networks on this skeleton to capture the equilibrium phenomena of opinions. Subsequently, a refine operation is used to reconstruct the features of the skeleton nodes back to the original nodes. This approach captures the equilibrium phenomena of opinion dynamics while ensuring that nodes remain distinguishable, thereby avoiding over-smoothing. In this manner, the gap between theoretical models of opinion dynamics and real-world opinion evolution is bridged.

In this paper, UniGO is proposed as a unified framework for modeling opinion evolution on graphs. The framework extracts supernodes from social media graphs through graph pooling, using information from the original nodes to construct a topological skeleton for the supernodes. A graph neural network is then employed to effectively model unified opinion dynamics on this skeleton. Finally, a refine module is used to lift the opinion information from the skeleton back to the original nodes, maintaining equilibrium phenomena while avoiding over-smoothing in large-scale networks. By constructing a unified opinion dynamics model, synthetic data with complex fusion rules is generated to enhance model performance. The main contributions are as follows:

(1) We propose a unified opinion dynamic to generate complex synthetic data, enhancing the model's ability to learn the complex intrinsic relationships between opinion formation and graph topology, while incorporating complex opinion fusion rules.

(2) We introduce UniGO, a graph nerual network model for modeling opinion evolution on graphs. By leveraging the coarsen-refine framework, we balance the equilibrium phenomena of opinion evolution while avoiding the over-smoothing problem commonly faced by graph neural networks.

(3) We conduct experiments on both synthetic and real-world network datasets.

## 2 RELATED WORKS

### 2.1 Opinion dynamics model

Opinion dynamics investigates the formation process of agent opinions in social systems over time by setting up fusion rules. The format of opinions, fusion rules, and underlying structures are pivotal components of opinion dynamics [18]. This paper primarily concentrates on dynamic models grounded in continuous opinion forms, which offer a closer alignment with the real-world social systems[5].

Different opinion dynamics models assume different fusion rules and study the phenomena of opinion formation based on these rules [42]. Based on the assumption of opinion assimilation, the DeGroot model [16] sets the fusion rule as a simple averaging form. Friedkin-Johnsen (FJ) model [21] introduces agent stubbornness to model situations where agent persist in their opinions in real-world scenarios. The Deffuant-Weisbuch (DW) [15] model and Hegselmann-Krause (HK) [24] model consider the phenomenon of bias assimilation [20], where agents tend to accept opinions they are inclined to believe, introducing confidence thresholds where agents can only interact with others whose opinions are within a certain range. Subsequent research delves into more intricate fusion rules, including randomness in opinion fusion [10, 17, 38], deceptive interactions in social networks [4], group influence [25], and so forth.

In recent years, many studies have increasingly focused on the influence of underlying topology on opinion dynamics [32, 34, 37, 41].

Unlike face-to-face offline social systems, agents in online social systems often only interact with a subset of agents. Through simple transformations, the above theoretical models can be effectively transferred to graph structures. These models utilize synthetic graphs and real-world graph data to study the impact of topologies on opinion dynamics.

These theoretical models have played a significant role in analyzing social systems. However, most opinion dynamics models can only capture certain aspects of opinion fusion in real-world scenarios and fail to model the complexity of opinion evolution. Integrating insights from opinion dynamics into data-driven models can effectively combine perspectives from both theory and empirical data.

## 2.2 Data-driven methods for modeling opinion dynamics

Benefiting from the data-driven paradigm, many researchers have studied learning the formation patterns of opinions from data. Some machine learning methods have recognized the importance of graph topology. The work [13] uses a linear model to capture transient changes in opinions to adapt to complex opinion scenarios. The work [9] extends the k-means method to social graphs, investigating the formation of opinion equilibrium from a community clustering perspective. AsLM [12] proposed a linear influence model that explores how to estimate the influence strength of links in social graphs by observing the opinions evolving over time at the nodes. Additionally, this work considers the limitations of the interaction range of nodes, indirectly introducing the underlying topology. SLANT [14] describes users' latent opinions as continuous-time stochastic processes, where each expression of an opinion by a user is a noisy estimate of their current latent opinion. SLANT+ [28] is a point-process-based framework for capturing the nonlinear dynamics of opinion fusion. It processes users' opinions and message timings as a temporal point process, influenced by the opinions and message timings of their neighbors. Some methods are inspired by the fusion rules in opinion dynamics. SINN [36] incorporates the idea of [26, 44] and uses opinion dynamics to assist model training.

These approaches partially integrate knowledge from opinion dynamics models but fail to fully incorporate this knowledge into the internal mechanisms of the model. Additionally, these methods do not explicitly incorporate graph topology into the opinion fusion process, which is inconsistent with the interaction patterns observed in large-scale social media. A unified opinion dynamics model is proposed to integrate interaction rules from various classical opinion dynamics models. Furthermore, this work delves into learning opinion dynamics on graphs using graph neural networks and applies them to real-world data, further bridging the gap between theoretical models of opinion dynamics and real-world applications.

## 3 BACKGROUND

This section introduces the concept of opinion dynamics on graphs, along with several significant theoretical models of opinion dynamics. Furthermore, the discussion covers the opinion fusion rules that each of these models focuses on. Then, in section 3.2, the connections between graph neural networks and opinion dynamics on graphs are explored, and the key challenges in learning opinion dynamics on graphs are summarized.

## 3.1 Opinion dynamics models on graphs

Opinion dynamics models describe the formation process of opinions in social systems. Classical opinion dynamics models are often formulated as difference equations to describe specific opinion fusion rules. The opinions of agents evolve over time steps, where the opinion at time $t + 1$ is based on the fusion of opinions within the agents at time $t$. With continuous iterations, the opinions of agents eventually converge to certain values, a process also known as equilibrium. By proposing more complex opinion fusion rules, opinion dynamics can effectively describe opinion formation phenomena in the real world, including consensus, polarization, and fragmentation.

In large-scale social systems in the real world, nodes can only interact with a small subset of others. Therefore, graph-based opinion dynamics are proposed to address such opinion formation constrained by underlying graph topology. Given a social graph $\mathcal{G} = (\mathcal{V}, \mathcal{E}, \mathcal{X})$, where $\mathcal{V}$ is the set of nodes with $|\mathcal{V}| = n$ representing the number of nodes, each node represents an agent. $\mathcal{E}$ is the set of edges, with $e_{ij} \in \mathcal{E}$ representing an edge between nodes $v_i$ and $v_j$ (such as friendships, follow relationships, etc.), and $\mathcal{X}$ represents the opinions held by each node.

*Node Attributes.* Some opinion dynamics models assume that nodes possess specific attributes that influence opinion fusion. For example, in the FJ model, each agent has a stubbornness coefficient, which represents the extent of the node's adherence to its own opinion.

$$x_i(t + 1) = (1 - \lambda_i)x_i(0) + \lambda_i \sum_{v_j \in \mathcal{N}(v_i)} w_{ij}x_j(t). \tag{1}$$

Here, $\lambda_i \in [0, 1]$ represents the weight assigned to social influences during opinion fusion, and $1 - \lambda_i$ is the weight assigned to the agent's initial opinion value, i.e., the stubbornness of the nodes towards their initial opinion $x_i(0)$.

*Edge Attributes.* Opinion dynamics based on confidence thresholds control the range of node interactions, essentially modifying the state of edges between nodes. For example, the HK model describes the influence of all neighbors' opinions within each agent's trust range. The condition for nodes to interact is that the difference in opinions between the two nodes must fall within a certain range, known as the confidence threshold $d$. Specifically, agent $v_i$ can interact with nodes whose opinion values lie within the interval $[x_i - d, x_i + d]$, which is referred to as the confidence interval of $v_i$. When $|x_i(t) - x_j(t)| < d$, interaction occurs.

$$x_i(t + 1) = \frac{1}{|\mathcal{N}_i'(t)|} \sum_{j \in \mathcal{N}_i'(t)} x_j(t). \tag{2}$$

Here, $\mathcal{N}_i'(t)$ represents the set of nodes within the confidence threshold in the neighborhood of $v_i$ at time $t$, including $v_i$ itself, and $|\mathcal{N}_i'(t)|$ represents the number of elements in set $\mathcal{N}_i'(t)$.

*Community Structure.* Building on the aforementioned work, some opinion dynamics models consider the influence of community structure on opinion fusion. For example, certain models assume that the stubbornness coefficients and confidence thresholds are structurally correlated, meaning that nodes in different communities within the graph may have distinct stubbornness coefficients and confidence thresholds.

$$\lambda_i = \lambda_c, d_i = d_c \quad \forall v_i \in C_k \tag{3}$$

Here, $C_k$ represents the $k$-th community, while $\lambda_c$ and $d_c$ denote the stubbornness coefficient and confidence threshold of nodes within the community, respectively.

*Noise.* In real social media, the formation of user opinions is often influenced by various sources of noise. This noise may originate from algorithmic biases or the uncertainties of user behavior. To better simulate these effects, some opinion dynamics models introduce noise terms. In noise-considering models of opinion fusion [38], individuals may randomly change their opinions based on a certain probability, rather than merely fusing the opinions of their neighbors. Such noise can introduce new collective behaviors, making the model's behavior more complex and realistic.

$$x_i(t+1) = \begin{cases} \text{Randomly select an } x \in [0, L] & \text{with probability } m \\ \frac{1}{|\mathcal{N}_i(t)|} \sum_{j \in \mathcal{N}_i(t)} x_j(t) & \text{with probability } 1 - m \end{cases} \tag{4}$$

Here, each individual has a probability $m$ to make a random jump, after which the new opinion $x_i(n + 1)$ is randomly selected within the entire range $[0, L]$.

## 3.2 Learning opinion dynamics on graphs

When we aim to learn opinion dynamics on graphs, graph neural networks (GNNs) naturally become the preferred method for capturing this inductive bias. In this subsection, we explore the connections and differences between GNNs and graph-based opinion dynamics, and describe the key challenges in learning opinion dynamics on graphs.

*Opinion updates and node representation updates.* Consider a typical graph neural network, where node representations are updated by aggregating the information from neighboring nodes. The $(k + 1)$-th layer can be defined as:

$$\mathbf{h}_i^{k+1} = AGG^{k+1} \left( \left\{ MSG^{k+1}(\mathbf{h}_j^k, \mathbf{h}_i^k) | v_j \in \mathcal{N}(v_i) \right\} \right), \tag{5}$$

where $AGG(\cdot)$ is the aggregation function at layer $k + 1$, $MSG(\cdot)$ is the message passing function at layer $k + 1$, $\mathcal{N}(v_i)$ is the neighborhood of node $v_i$, and $\mathbf{h}_i^{k+1}$ is the representation of node $v_i$ at layer $k + 1$. Similarly, opinion dynamics on graphs can be written in a similar form:

$$x_i(t + 1) = FUS \left( x_i(t), x_j(t) | j \in \mathcal{N}(v_i) \right), \tag{6}$$

where $x_i$ represents the opinion of $v_i$ at time $t + 1$, and $FUS(\cdot)$ is the fusion function, computed based on the opinions of the node and its neighbors at time $t$. This formal consistency indicates the close connection between GNNs and opinion dynamics on graphs.

From the perspective of GNNs, each iteration of opinion dynamics allows nodes to receive information from an additional hop of their neighborhood. Graph-based opinion dynamics often require dozens to hundreds of steps to reach equilibrium, meaning each node eventually receives global information. Similarly, from the perspective of opinion dynamics, each layer of network updates can be seen as an iterative process in one time step. This also implies that the 'time steps' in opinion dynamics do not correspond to the real-world concept of time, allowing the use of static graph neural networks for research.

*Equilibrium and over-smoothing.* Equilibrium is a crucial aspect studied in opinion dynamics theoretical models, referring to the state where, after multiple iterations, the opinions of nodes no longer change with each time step. In GNNs, over-smoothing refers to the phenomenon where, as the number of network layers increases, node representations become increasingly similar, eventually becoming indistinguishable and thus making it difficult to differentiate between different nodes [35]. The similarity between the two lies in the convergence of node states (representations). The fundamental difference is that equilibrium in opinion dynamics is related to the graph structure and fusion rules, reflecting the characteristics of the dynamic system, whereas over-smoothing is an undesirable effect during the training process.

The greatest challenge in using GNNs, which update based on local information, to learn opinion dynamics on graphs with global structural properties is learning a pattern of node convergence while avoiding the issue of over-smoothing.

## 4 METHOD

To address the aforementioned issues, we propose UniGO, a unified framework based on graph neural networks for modeling opinion evolution on graphs. As illustrated in fig. 2, UniGO is trained on synthetic datasets generated under unified opinion dynamics and tested on real-world datasets. The data synthesis module consists of two components: graph construction and dynamics construction. Graph construction generates graph structures using various random graph generation methods, while dynamics construction generates evolutionary data based on unified opinion dynamics on the graph structure.

The UniGO model comprises three parts: coarsening, dynamics simulation, and refinement. The coarsening module uses graph pooling methods to generate a skeleton of the original graph and construct aggregated representations of the supernodes. Then, in the dynamics simulation module, graph neural networks are employed to simulate the dynamics evolution on the supernodes. Finally, in the refinement module, the supernode representations are refined back to the original nodes to complete the simulation of opinion evolution on the graph. The coarsen-refine architecture allows learning the equilibrium results of opinion dynamics on graphs while avoiding over-smoothing.

## 4.1 Problem definition

Given a graph $\mathcal{G} = (\mathcal{V}, \mathcal{E}, \mathbf{X}_l)$, where $\mathcal{V}$ is the set of nodes, $\mathcal{E}$ is the set of edges, and $\mathbf{X}_l \in \mathbb{R}^{n \times t_l}$ represents the opinions of $n$ nodes at the initial $t_l$ time steps, with values in the range $[0, 1]$. The objective is to learn a function $F$ to predict the node opinions for the subsequent $t_h$ time steps, $\mathbf{X}_h \in \mathbb{R}^{n \times t_h}$.

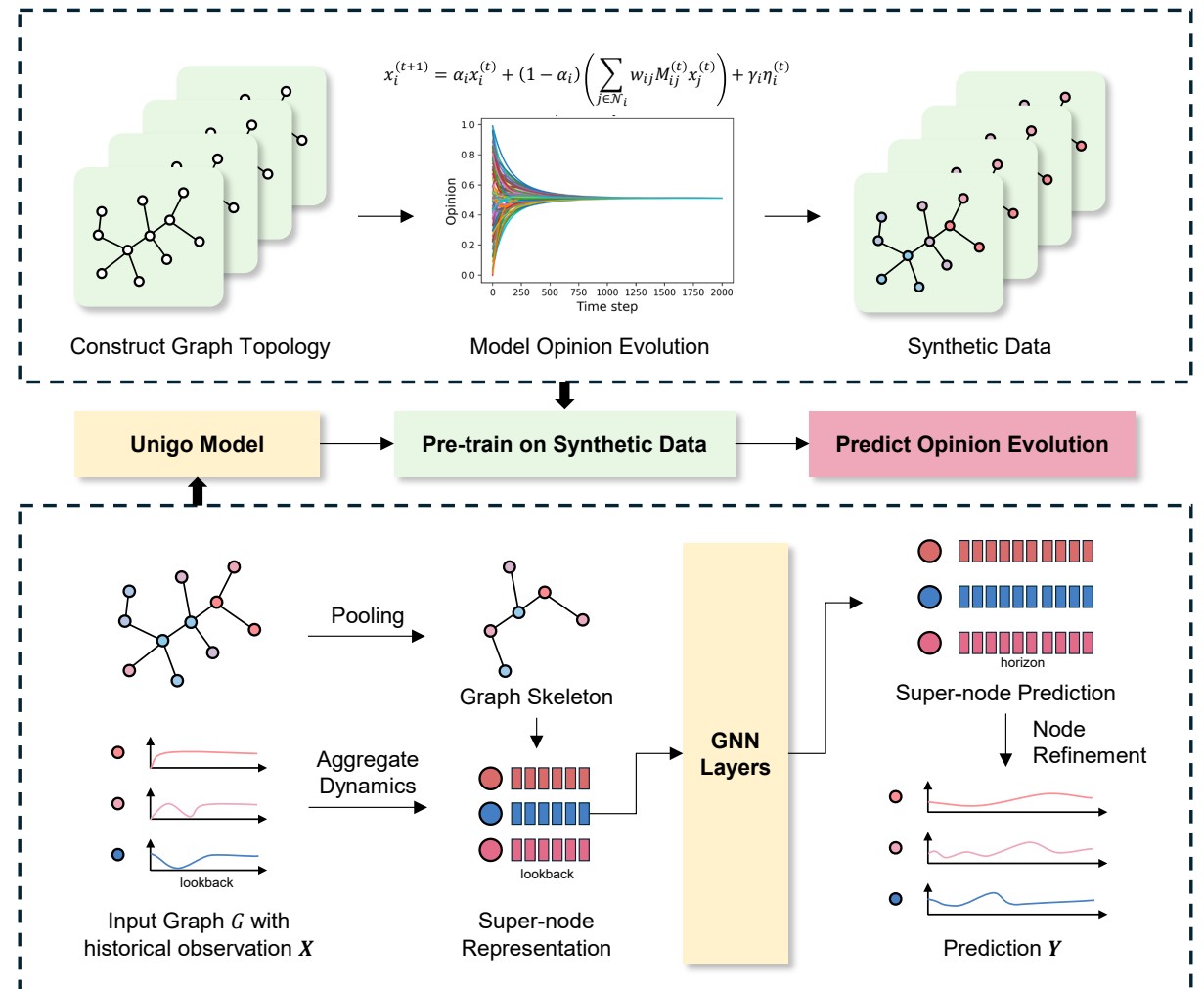

$$x_i^{(t+1)} = \alpha_i x_i^{(t)} + (1 - \alpha_i)\left(\sum_{j \in \mathcal{N}_i} w_{ij} M_{ij}^{(t)} x_j^{(t)}\right) + \gamma_i \eta_i^{(t)}$$

Figure 2: The framework of MOGO.

## 4.2 Data Synthesis Module

The data synthesis module generates synthetic datasets that include various graph structures and opinion dynamics processes, addressing the scarcity of real-world opinion evolution data. Specifically, the dataset consists of two components: graph construction and dynamics construction. The graph construction component includes several random graph generation methods, such as ER random graphs, WS small-world graphs, and BA scale-free graphs[3, 6, 45], with each node receiving an initial opinion randomly. In the dynamics construction component, the unified opinion dynamics runs on the random graphs to generate the opinion evolution process:

$$x_i^{(t+1)} = \alpha_i x_0^{(t)} + (1 - \alpha_i)\left(\sum_{j \in \mathcal{N}_i} w_{ij} M_{ij}^{(t)} x_j^{(t)}\right) + \gamma_i \eta_i^{(t)}. \quad (7)$$

Here, $x_0^{(t)}$ represents the initial opinion of node $i$ . $\alpha_i$ is the stubbornness coefficient of node $i$, $\mathcal{N}_i$ represents the set of neighboring nodes of node $i$, $w_{ij}$ is the weight of the edge between nodes $i$ and $j$, and $\eta_i^{(t)}$ is the noise for node $i$ at time $t$. $M_{ij}^{(t)}$ indicates whether nodes $i$ and $j$ can interact in terms of opinion exchange, and is defined as follows:

$$M_{ij}^{(t)} = \begin{cases} 1, & \text{if } |x_i^{(t)} - x_j^{(t)}| \le d_i \\ 0, & \text{otherwise.} \end{cases} \quad (8)$$

Here, $d_i$ is the confidence threshold of node $i$. Nodes $i$ and $j$ interact if the difference in their opinions is less than $d_i$.

In this way, traditional dynamics models can be effectively combined. For instance, when $w_{ij} M_{ij}^{(t)} = 1$ and $\gamma_i \eta_i^{(t)} = 0$, the model reduces to the FJ model. If the stubbornness coefficients and confidence thresholds of nodes vary according to the community structure, the model represents an FJ and HK model that considers graph structure. Furthermore, by designing different noise functions, the model can flexibly simulate the process of opinion evolution under varying noise conditions.

More importantly, this framework comprehensively incorporates different opinion fusion rules from various dynamics models, generating more realistic synthetic datasets for opinion dynamics. Although this framework cannot analyze equilibrium states and other properties in the traditional opinion dynamics paradigm, it can be used to train graph neural network-based models for predicting opinion evolution. Details of the data synthesis approach are provided in the appendix A.1.

### 4.3 UniGO Model

The UniGO model, under a coarsen-refine mechanism, uses graph neural networks to simulate the dynamics evolution on supernodes. The coarsening module coarsens the original graph through pooling methods to obtain the skeleton and construct aggregated representations of the skeleton supernodes. The dynamics evolution is simulated using a mean-aggregating graph neural network. The refinement module refines the supernode representations back to the original nodes to complete the simulation of opinion evolution on the graph.

*Coarsening Module.* Graph pooling methods are used to coarsen the original negraphtwork, obtaining the skeleton and constructing aggregated representations of the skeleton supernodes. First, a graph neural network is used to aggregate the dynamic representations of the nodes:

$$\mathbf{x}_i^{k+1} = AGG^{k+1}\left(\left\{MSG^{k+1}(\mathbf{x}_j^k, \mathbf{x}_i^k)|v_j \in \mathcal{N}(v_i)\right\}\right), \quad (9)$$

here, $AGG(\cdot)$ is the aggregation function at layer $k+1$, $MSG(\cdot)$ is the dynamic representation function at layer $k+1$, and $\mathcal{N}(v_i)$ represents the set of neighboring nodes of node $v_i$. The initial node feature $\mathbf{x}_i^{(0)}$ corresponds to the opinions from the first $t_l$ time steps.

Subsequently, following [1], the coarsened node representations are learned based on soft clustering. The core algorithm of this pooling layer is based on calculating distances from nodes to cluster centers and performing soft assignments. For each node $\mathbf{x}_i$ and cluster center $\mathbf{k}_j^{(h)}$, the distance $d_{ij}$ is computed using the Euclidean distance, with a temperature parameter $\tau$ introduced to adjust the influence of the distance:

$$d_{ij}^{(h)} = \left(1 + \frac{\|\mathbf{x}_i - \mathbf{k}_j^{(h)}\|^2}{\tau}\right)^{-\frac{1+\tau}{2}}, \quad (10)$$

here, $\mathbf{x}_i \in \mathbb{R}^F$ represents the feature vector of node $i$, and $\mathbf{k}_j^{(h)} \in \mathbb{R}^F$ represents the feature vector of cluster center $j$ for the $h$-th head. $\tau$ is a temperature parameter that controls the influence of the distance. The probability that node $\mathbf{x}_i$ belongs to different clusters $j$ is calculated by normalizing the distances. The probability $S_{ij}^{(h)}$ of each node being assigned to different clusters is expressed as:

$$S_{ij}^{(h)} = \frac{d_{ij}^{(h)}}{\sum_{k=1}^K d_{ik}^{(h)}}, \quad (11)$$

here, $S_{ij}^{(h)}$ represents the probability that node $i$ is assigned to cluster $j$. In the multi-head attention mechanism, the final soft assignment matrix $\mathbf{S}$ is further processed through a convolution operation:

$$\mathbf{S} = \text{softmax}(\text{Conv2d}(\|_{h=1}^H \mathbf{S}^{(h)})) \in \mathbb{R}^{N \times K}. \quad (12)$$

Finally, the node features $\mathbf{X}$ are weighted and combined according to the assignment matrix $\mathbf{S}$, resulting in the new node features $\mathbf{H}$:

$$\mathbf{H} = \mathbf{S}^\top \mathbf{X} \mathbf{W} \in \mathbb{R}^{K \times F'} \quad (13)$$

Here, $\mathbf{X} \in \mathbb{R}^{N \times F}$ represents the input node feature matrix, and $\mathbf{W} \in \mathbb{R}^{F \times F'}$ is the linear transformation matrix used to map the node features to the output space.

*Dynamics Evolution Module.* A mean-aggregating graph neural network is used to simulate the dynamics evolution process. Specifically, a graph neural network is employed to compute the dynamics evolution for each node, and these dynamics are then used to update the node features. A weighted mean aggregation approach is selected, with the specific formula given as follows:

$$\mathbf{h}_i^{k+1} = \mathbf{W}_1 \mathbf{h}_i^k + \mathbf{W}_2 \cdot \text{mean}_{j \in \mathcal{N}(i)} \mathbf{h}_j^k, \quad (14)$$

Here, $\mathbf{W}_1$ and $\mathbf{W}_2$ are learnable weight matrices. Finally, a linear layer is used to expand the supernodes to $t_h$ time steps:

$$\mathbf{Z} = \mathbf{H}^K \mathbf{W}_e + \mathbf{b}_e, \quad (15)$$

here, $K$ represents the number of layers in the graph neural network. $\mathbf{H}^K$ is the evolved representation of the supernodes, $\mathbf{W}_e \in \mathbb{R}^{t_l \times t_h}$ is the weight matrix, and $\mathbf{b}_e \in \mathbb{R}^{t_h}$ is the bias vector. Through this linear transformation, the time dimension of the supernodes is expanded from $t_l$ to $t_h$, resulting in $\mathbf{Z} \in \mathbb{R}^{K \times t_h}$.

*Refinement Module.* A refiner is used to refine the supernode representations back to the original nodes. First, the supernode representations are restored to the original nodes through the assignment matrix $S$:

$$\mathbf{Z}' = \mathbf{S}\mathbf{Z} \quad (16)$$

Here, $\mathbf{S} \in \mathbb{R}^{N \times K}$ is the assignment matrix, and $\mathbf{Z} \in \mathbb{R}^{K \times t_h}$ represents the evolved supernode representations. Subsequently, the original node representations $\mathbf{X}$ and the restored representations $\mathbf{Z}'$ are passed through a shared refiner and concatenated to obtain the final node representations:

$$\mathbf{Y} = \sigma((\mathbf{X}\mathbf{W}_x || \mathbf{Z}'\mathbf{W}_z)\mathbf{W}_y) \quad (17)$$

Here, $\mathbf{W}_x$ and $\mathbf{W}_z$ are the weight matrices corresponding to $\mathbf{X}$ and $\mathbf{Z}'$, respectively. $\mathbf{W}_y$ is the weight matrix of the refiner, $\sigma$ is the activation function, and $\mathbf{Y}$ represents the final model prediction result.

### 4.4 Training

The training loss of the model consists of two parts: the KL divergence of the assignment matrix and the mean squared error (MSE) of the node opinion prediction. The loss function is formulated as follows:

$$\mathcal{L} = \lambda \mathcal{L}_{KL} + \mathcal{L}_{MSE} \quad (18)$$

Here, $\mathcal{L}_{KL}$ represents the KL divergence of the assignment matrix, $\lambda$ is the weight of the KL divergence, and $\mathcal{L}_{MSE}$ represents the mean squared error (MSE) of the node opinion prediction.

## 5 EXPERIMENTS

This section comprehensively evaluates UniGO's performance on both synthetic and real-world datasets. First, we introduce the synthetic and real-world datasets used in the experiments. In section 5.2, the evaluation focuses on UniGO's ability to fit opinion evolution on synthetic datasets. In section 5.3, the evaluation covers UniGO's ability to predict opinion formation on real-world datasets. The findings show that pretrained models on synthetic datasets can effectively predict opinions in real-world datasets, providing a viable paradigm for applying opinion dynamics to large-scale real-world scenarios.

### 5.1 Experimental Setup

*Datasets.* The synthetic dataset UniSyn is generated using the unified opinion dynamics. This synthetic dataset includes three types of random graph structures, and the dynamics involve various parameter combinations. The details of the synthetic dataset are presented in Table 1, with the detailed construction process provided in appendix A.1.

A real-world dataset related to the Delhi elections [13] and four other datasets obtained from social media were used: Rumor, Food Safety, COVID, and U.S. Election. Corporate and bot accounts as well as irrelevant information were filtered out. The criteria for identifying active users varied by topic: the threshold for the U.S. election dataset was 50, for the food safety dataset it was 8, for the Israel-Palestine conflict rumor dataset it was 8, and for the COVID dataset it was 100. Edges were constructed based on retweet or comment relationships, and the largest connected component was retained. Detailed information on the real-world datasets is provided in appendix A.2.

**Table 1: Synthetic datasets information.**

| Dataset | Number of Graphs | Average Nodes | Average Edges |
|---------|------------------|---------------|---------------|
| UniSyn  | 3878             | 1652.4        | 16336.11      |

*Baselines.* In this experiment, there is a key difference between synthetic and real-world data: the graph structure in the synthetic dataset is not fixed, whereas the graph structure in the real-world dataset is fixed. Therefore, for the synthetic dataset, baselines that can handle flexible graph structures are selected, including **GCN** [27], **GAT** [43], **GIN** [47], **GraphSAGE** [23], **SGC** [46], and **ODNET** [33]. For the real-world dataset, several state-of-the-art dynamics prediction models are added, including **NCDN** [48], **SINN** [36], and **DiskNet** [31]. Detailed information about these models is provided in appendix A.3.

### 5.2 Learning opinion formation on synthetic datasets

*Experimental setup.* In this subsection, UniGO's ability to learn opinion dynamics on UniSyn is evaluated. In this experiment, $t_l$ is set to 10, and $t_h$ is set to 90. For each dataset, only the first 100 time steps are used for the experiment. Unlike using a sliding window to construct multiple datasets, each graph structure and its corresponding dynamics evolution are sampled only once to learn the impact of graph structure and dynamics on evolution. For each graph, the mean squared error (MSE) of all nodes and the mean Wasserstein distance (MWD) are used to evaluate the model's performance.

**Table 2: Results on synthetic datasets ($\times 10^{-3}$).**

| Methods | MSE | MWD |
|---------|-----|-----|
| GCN | $26.5 \pm 3.2$ | $153.6 \pm 2.3$ |
| GAT | $15.3 \pm 3.3$ | $102.6 \pm 5.6$ |
| GIN | $18.6 \pm 2.6$ | $90.3 \pm 4.2$ |
| GraphSage | $7.1 \pm 1.2$ | $64.7 \pm 1.2$ |
| SGC | $20.7 \pm 1.8$ | $123.4 \pm 2.7$ |
| ODNET | $25.3 \pm 3.4$ | $177.1 \pm 2.5$ |
| UniGO-c | $8.0 \pm 1.8$ | $66.8 \pm 2.2$ |
| UniGO | $\mathbf{3.2 \pm 0.8}$ | $\mathbf{27 \pm 1.5}$ |

*Results.* Table table 2 shows the MSE and MWD for different methods on the synthetic dataset. Bold text indicates the best result. UniGO-c refers to UniGO without using the coarsen-refine architecture. UniGO effectively captures the opinion evolution process on synthetic data. Although ODNET adopts the confidence interval assumption from opinion dynamics, it fails to capture changes brought about by other opinion fusion rules, resulting in inferior performance compared to some common graph neural networks, such as GAT and GIN. It is noted that GraphSage's performance is second only to UniGO. One possible reason is that both UniGO-c (without the coarsen-refine architecture) and GraphSage use mean aggregation, but GraphSage employs a sampling technique when aggregating neighbor information. This allows GraphSage to better avoid overfitting during training, resulting in improved performance. The comparison between UniGO and UniGO-c also demonstrates the necessity of the coarsen-refine architecture.

### 5.3 Learning opinion formation on real datasets

*Experimental Setup.* Furthermore, the generalization ability and opinion formation prediction capability of UniGO are analyzed on real-world datasets. For real datasets, a sliding window approach is used to construct the dataset. Similarly, $t_l$ is set to 10, and $t_h$ is set to 90. Unlike synthetic datasets, the graph structure in real datasets is fixed, allowing multiple datasets to be constructed using the sliding window approach. This also makes the data suitable for most baselines. The dataset is split into training, validation, and test sets in a ratio of 0.7, 0.1, and 0.2, respectively.

Additionally, to validate the effectiveness of UniGO under a pretrain-finetune framework, the UniGO model trained on the

**Table 3: Results on real datasets ($\times 10^{-3}$).**

| Method | Delhi Election | | U.S. Election | | Rumor | | Food Safety | | COVID | |
|---|---|---|---|---|---|---|---|---|---|---|
| | MSE | MWD | MSE | MWD | MSE | MWD | MSE | MWD | MSE | MWD |
| GNN | 10.6 ± 1.8 | 80.5 ± 0.5 | 11.2 ± 1.2 | 82.6 ± 1.9 | 35.6 ± 0.7 | 125.6 ± 1.4 | 15.6 ± 0.3 | 95.2 ± 1.6 | 9.6 ± 1.1 | 50.7 ± 0.8 |
| GAT | 10.5 ± 0.4 | 79.7 ± 1.7 | 11.2 ± 0.9 | 81.9 ± 0.6 | 36.9 ± 1.5 | 123.5 ± 0.2 | 16.8 ± 1.8 | 89.5 ± 1.3 | 13.5 ± 0.7 | 79.4 ± 1.9 |
| GIN | 12.4 ± 1.6 | 85.7 ± 0.8 | 9.5 ± 1.3 | 84.7 ± 1.1 | 37.8 ± 0.5 | 115.6 ± 1.7 | 20.8 ± 0.9 | 87.9 ± 0.4 | 12.1 ± 1.4 | 48.6 ± 0.6 |
| GraphSage | 8.6 ± 0.7 | 54.6 ± 1.5 | 7.4 ± 0.3 | 72.3 ± 1.8 | 21.5 ± 1.2 | 100.2 ± 0.9 | 12.3 ± 1.6 | 70.4 ± 0.5 | 8.6 ± 1.0 | 30.9 ± 1.3 |
| ODNET | 15.6 ± 1.1 | 90.5 ± 0.6 | 11.6 ± 1.9 | 78.2 ± 0.8 | 31.8 ± 0.4 | 105.4 ± 1.5 | 17.0 ± 1.2 | 75.9 ± 1.7 | 13.9 ± 0.3 | 38.1 ± 0.9 |
| SINN | 6.5 ± 1.4 | 42.1 ± 0.7 | 6.4 ± 1.0 | 39.0 ± 1.6 | 19.8 ± 0.8 | 97.7 ± 1.3 | 9.5 ± 0.5 | 59.5 ± 1.9 | 5.9 ± 1.1 | 33.4 ± 0.2 |
| NCDN | 10.6 ± 0.9 | 78.6 ± 1.2 | 9.7 ± 0.6 | 59.5 ± 1.4 | 18.7 ± 1.8 | 81.7 ± 0.3 | 10.0 ± 1.5 | 74.3 ± 0.7 | 4.6 ± 1.0 | 22.2 ± 1.6 |
| DiskNet | 10.1 ± 0.5 | 76.4 ± 1.8 | 10.6 ± 1.3 | 74.5 ± 0.4 | 16.8 ± 1.7 | 72.4 ± 0.9 | 8.4 ± 1.1 | 42.9 ± 1.5 | 5.2 ± 0.6 | 42.2 ± 1.2 |
| UniGO$_s$ | 7.4 ± 1.6 | 50.4 ± 0.8 | 6.8 ± 1.2 | 43.5 ± 0.5 | **8.7 ± 1.9** | **38.6± 1.0** | 7.4 ± 0.3 | 30.8 ± 1.7 | 4.2 ± 0.9 | 31.8 ± 1.4 |
| UniGO | **4.0 ± 1.1** | **28.9 ± 0.7** | **5.2 ± 1.5** | **30.5 ± 1.8** | 11.3 ± 0.4 | 42.2 ± 1.3 | 7.6 ± 0.6 | 35.9 ± 1.0 | **4.0 ± 1.9** | **25.7 ± 0.2** |

UniSyn dataset is directly tested on the real-world dataset to evaluate its generalization capability. UniGO$_s$ represents the UniGO model trained on the UniSyn dataset.

*Results.* Table table 3 shows the MSE and MWD for different methods on the real-world dataset. Bold text indicates the best result, and underline indicates the second-best result. From the experimental results, it can be seen that UniGO performs well on smaller datasets, consistently achieving the best results, while UniGO$_s$ performs better on relatively larger datasets. This is primarily because UniGO$_s$ was not fine-tuned on the real-world data but directly tested on it, which led to suboptimal performance on smaller datasets. SINN, as an opinion dynamics-informed neural network model, performs well on most tasks, demonstrating the advantages of incorporating opinion dynamics into opinion prediction.

It is also observed that the time span and size of the dataset have a certain impact on the results. For the Delhi Election dataset, which has a shorter time span and smaller data volume, UniGO$_s$ may have overfitted, resulting in less ideal performance. Almost all methods perform relatively worse on larger datasets. However, UniGO$_s$ shows more stable performance. Similarly, under the coarsen-refine mechanism, the Graph Neural ODE-based model DiskNet performs worse than SINN on smaller datasets, possibly due to overfitting caused by the more complex Graph Neural ODE model compared to the MLP used in SINN.

Overall, the UniGO model shows good performance across different datasets. The UniGO model pretrained on synthetic data demonstrates stable performance across datasets of different sizes, also showing good generalization ability.

## 6 CONCLUSION

This paper proposes UniGO, a unified opinion dynamics-based model for predicting opinion evolution on graphs, aiming to simulate and predict dynamic opinion formation. The unified opinion dynamics for neural network learning considers various opinion fusion rules to overcome the limitations of traditional opinion dynamics, which cannot simulate complex opinion fusion processes. By pretraining on synthetic datasets, UniGO effectively generalizes to real-world datasets, providing a viable paradigm for applying opinion dynamics in practical applications. This approach addresses challenges such as sparse real data, simplistic theoretical model assumptions, and over-smoothing in graph neural networks, combining the interpretability of theoretical models with the fitting ability of data-driven models, and offering a new direction for future research. Experimental results show that UniGO achieves good performance in opinion prediction tasks on both synthetic and real datasets.

Limitations: (1)Limitations of the Unified Opinion Dynamics. Although the proposed unified opinion dynamics can be applied to various opinion fusion rules, it still has certain limitations, such as the inability to handle coupled opinion dynamics or changes in the underlying graph structure over time. Future work will explore more general representations of opinion dynamics and apply them to more complex systems. (2) Limitations of the Real Datasets. Larger-scale and more diverse real-world datasets could further verify the generalization capability of the model. Additionally, all real datasets used are based on text data, making the accurate transformation of text into opinion data an area worth investigating. Future research will focus on developing more accurate methods for generating opinion data from textual information.

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

# A  EXPERIMENTAL DETAILS

## A.1  Build Synthetic Datasets

BA, ER, and WS graph structures are used to generate random networks. The network size is set as $n \in \{500, 1000, 2000, 3000\}$. For BA random graphs, the average degree is $k \in \{2, 3, 4, 5\}$. For ER random graphs, the probability of any two nodes being connected is $p \in \{0.1, 0.2, 0.3\}$. For WS random graphs, the average degree is $k \in \{4, 6, 8\}$, and the probability of any two nodes being connected is $p \in \{0.1, 0.2, 0.3\}$. For each set of parameters, five random network structures are generated to run the unified opinion dynamics model.

For the opinion dynamics model, the stubbornness parameter is set as $\alpha \in \{0.3, 0.4, 0.5, 0.6, 0.7\}$, the confidence threshold is $\epsilon \in \{0.1, 0.2, 0.3, 0.4\}$, and the noise parameter is $\sigma \in \{0.05, 0.1, 0.15, 0.2, 0.25\}$. For models where parameters are determined based on network structure, the Louvain method is used for community detection, with the number of communities set to 5. For all parameters that need to be randomly selected, three levels are set: [0.1, 0.3], [0.2, 0.4], and [0.3, 0.5]. On each graph structure, multiple opinion dynamics models are run, provided there are no conflicting parameters. Opinion dynamics are calculated using the Euler method, running for 200 time steps. If the average opinion change of nodes over 10 continuous time steps is within $1 \times 10^{-4}$, the model is considered to

**Table 4: Statistics of the real-world datasets.**

| datasets | nodes | edges | message | time span | time step |
|---|---|---|---|---|---|
| Delhi Election | 548 | 5271 | 20026 | December 9, 2013 - December 15, 2013 | 280 |
| U.S. Election | 526 | 2482 | 10883 | December 1, 2023 - March 5, 2024 | 384 |
| Rumor | 10695 | 89248 | 201365 | October 7, 2023 - October 19, 2023 | 260 |
| Food Safety | 1390 | 4253 | 8506 | May 30, 2023 - July 14, 2023 | 276 |
| COVID | 894 | 21143 | 102635 | January 1, 2020 - March 31, 2020 | 364 |

have converged. Finally, data with convergence steps greater than 80 are selected to generate the synthetic dataset.

## A.2 Real datasets

We used a real dataset related to the Delhi elections [13] and four other datasets obtained from social media. We filtered out corporate and bot accounts as well as irrelevant information. The criteria for identifying active users varied by topic. The threshold for the U.S. election dataset was 50, for the food safety dataset it was 8, for the Israel-Palestine conflict rumor dataset it was 8, and for the COVID dataset it was 100. We constructed edges based on retweet or comment relationships and retained the largest connected component.

We used a combination of manual annotation and large language models [11] to label the opinions of the nodes. In the election dataset, user opinions corresponded to the political party they supported. In the food safety dataset, social network users discussed whether a food safety incident had occurred. In the rumor dataset, users chose whether or not to believe the rumors. In the COVID dataset, user opinions reflected their optimism or pessimism about the pandemic. The table table 4 shows the information of these real datasets.

## A.3 Baselines

In the experiments, in addition to the commonly chosen graph neural networks—**GCN** [27], **GAT** [43], **GIN** [47], **SGC** [46] and **GraphSAGE** [23]—the following baselines are used for comparison:

- **ODNET** [33], a graph neural network inspired by HK opinion dynamics. It combines opinion dynamics with the message-passing mechanism in neural networks, using the concept of confidence thresholds to adjust influence weights between nodes.
- **NCDN** [48], a continuous-time graph neural network model. Unlike traditional neural networks with discrete layers, NDCN processes GNN layers in continuous time using numerical integration, thereby capturing the dynamic changes in node states on the graph.
- **SINN** [36], a sociology-driven neural network that translates opinion dynamics theoretical models from sociology into ordinary differential equations (ODEs), using them as constraints within the neural network.
- **DiskNet** [31], a model that incorporates the renormalization group concept to construct a coarsening framework for networks, and uses graph neural ODEs to learn on the coarsened network.

## A.4 Additional experimental setups

In this subsection, we introduce the detailed additional settings used in our experiments. The experiments ran on a Linux server with four NVIDIA RTX A6000 GPUs and 128 GB memory. For UniGO, the hyperparameters include the learning rate, the number of supernodes in graph pooling, the number of layers in the graph neural network for simulating dynamics, the weight of the KL divergence, and dropout. The learning rate of Adam was set to $\{0.01, 0.001, 0.0001\}$. The number of supernodes in pooling was set to $\{10, 20, 30, 40, 50, 60, 70, 80, 90, 100\}$, and the number of layers was set to $\{2, 3, 4, 5, 6\}$. The weight of KL divergence was set to $\{0, 0.2, 0.4, 0.6, 0.8, 1, 2, 5, 10\}$, and the dropout was set to $\{0.1, 0.2, 0.25\}$. For the experiments in section 5.2 and section 5.3, UniGO and other graph neural networks use a 3-layer setup, with the number of supernodes set to 50. In graph neural ODE-based methods, the discrete time step $dt$ is set to 0.1. SINN [36] is a model that does not directly consider network topology but models opinion dynamics as ODE constraints. Its input is data streams, so the original setup was followed.

