# OpenReview forum: "UniGO: A Unified Graph Neural Network for Modeling Opinion Dynamics on Graphs"
_ACM.org/TheWebConf/2025/Conference — WWW 2025 Oral_

### Official Review · Reviewer_C4RP · 2024-11-08

**Novelty:** 4
**Technical Quality:** 4

**Review:**

This paper studies the node state prediction in time-series. The authors seem to re-devise two modules from existing works. The main workflow and motivation are similar with DiskNet [31]. Overall, there are some areas need clarification and further design, particularly in the experimental setup and detailed explanation of contributions.

---
Pros:

1. The node state prediction is an interesting problem.

2. It makes sense to try to unify multiple opinion dynamic rules.

---

Cons:

1. The novelty of the contributions is somewhat unclear, as the proposed UniGO shares similarities with existing model DiskNet [31], which proposes the _coarsen-refine architecture_ along with the _graph skeleton_ and _super-node_. The authors should further clarify their own contributions and explain how their model differs significantly from existing models. If it is just an adjustment of usage, it may not be novel enough.

2. The motivation of unified model is not convincing enough. For unifying multiple assumptions within a single model, the authors should discuss how various fusion rules may interact, and whether conflicts or redundancies among them could influence the model performance. Currently, the paper lacks both theoretical analysis and empirical validation for this.

3. The authors should provide a theoretical analysis, as previous works done, particularly concerning the implications of training on synthetic graphs. It is essential to mathematically evaluate whether synthetic data can accurately represent real-world diffusion and dynamics. Such an analysis would strengthen the validity of using synthetic data in the model.

4. Experimental evaluation is not sufficient enough. The evaluation primarily presents overall prediction performance without in-depth analysis for key components. The results of model variants (such as UniGO_s and UniGO_c) are not well explained. More deatiled ablation experiments, particularly focused on synthetic graph construction, would improve understanding of each component’s contribution.

5. Overstatement of Scalability. The authors claim that the model is scalable for large-scale applications (e.g., Lines 24, 267, 711, and so on), but the largest dataset tested includes **only 10,695 nodes**, which does not constitute a large-scale graph. The authors should test the model on larger datasets to support this claim.

6. In Figure 2, the variant UniGO_s is presented, but it is not clearly labeled, which could be confusing for readers. The caption for this figure also contains typos. Ensuring clear labeling and captions for all figures will improve the manuscript’s readability.

**Questions:**

Some questions are discussed in the above Review comments.

Q1: The dynamics evolution is still modeled using a mean-aggregating GNN. Therefore, how does the model address the potential over-smoothing effect during training? In my opinion, the coarsen-refine architecture contributes primarily to mitigating this effect. However, it is proposed by the baseline DiskNet [31]. Could you please further explain the _essential_ differences between them, esplically the motivation and workflow?

Q2: The claim in Lines 143-146 regarding the effectiveness of combining multiple assumptions could benefit from further clarification. Could the authors include ablation experiments and parameter analysis to quantify the impact of different opinion fusion rules? I think this is crucial to a framework that aims to be unified.

Q3: Could the authors provide an analysis of time and space complexity for potential practical applications?

**Reviewer Confidence:**

4: The reviewer is certain that the evaluation is correct and very familiar with the relevant literature

**Scope:**

4: The work is relevant to the Web and to the track, and is of broad interest to the community

---

### Official Review · Reviewer_NvnW · 2024-12-02

**Novelty:** 5
**Technical Quality:** 4

**Review:**

This paper introduces a novel framework for modelling opinion evolution on graphs. The authors make two main contributions: a data synthesis model which generates synthetic graph data that can be used to train and evaluate opinion dynamics models. A coarsen-refine based graph neural network based model which aims to balance the equilibrium phenomena of opinion evolution while trying to avoid over-smoothing. The authors evaluate their GNN model against baseline on both synthetic data generated by their synthesis model, and on real world data.

# Strengths:
- The writing of the manuscript was clear and was overall pleasant to read.
- The application of the coarsen-refine framework to opinion dynamics modelling is interesting and novel.
- The work tested against a wide range of baseline models.
- The empirical results are impressive, with the proposed model outperforming the baselines by a significant margin on the synthetic dataset. The proposed model also outperformed existing baselines in real world datasets in the majority of cases.

# Weaknesses
- I am not sure where and how the multi-head attention mechanism comes into the model. It is not clear to me what h in this context is, is this still referring to the timesteps defined in 4.1?. Formula 12 is confusing and I was not able to understand what was being done. It seems convolution is being done across the concatenated set of assignment vectors for each head. The formula itself is also self-dependent on S which further contributes to the confusion. It is not to clear to me how the softmax is applied here (I assume along N?).

- I found the discussion on the real world datasets used for the experiments to be lacking in detail. The authors state that four other datasets were obtained from social media, but do not mention which social media platform or how this data was obtained. It would appear that at X/Twitter is in some way relevant given that edges are stated to be constructed based on retweets but this should be explicitly stated. There is no discussion regarding what constitutes a topic and how topics were identified. There is also no discussion on how annotations of the opinions were done in the main text, with the discussion in the appendix regarding annotation being very vague.

- Many of the references link to arxiv preprints despite the literature being published for many years (e.g. [27], [43], and [47] to list a few). Please fix these references to include the journals in the citations for already published works.

- In table 3, NCDN appears to have the best performance for MWD with 22.2 +- 1.6

**Questions:**

How does the multi-headed attention mechanism come into the model?

The authors state that for the real-world datasets “Corporate and bot accounts as well as irrelevant information were filtered out.” Could the authors please provide more details regarding this process, specifically regarding how bot accounts were identified, what constituted irrelevant information, and how this information was identified.

What is MOGO in Figure 2?

Slightly nitpicky but how does figure 1B show the similarities between the embedding update relates to the opinion update in the figure illustrations. Both the opinion update and embedding update show the same image, with the only difference being the formula. While I can understand intuitively that similarities between the two updates exist, it is unclear to me how this is shown by the figure.

**Reviewer Confidence:**

2: The reviewer is willing to defend the evaluation, but it is likely that the reviewer did not understand parts of the paper

**Scope:**

4: The work is relevant to the Web and to the track, and is of broad interest to the community

---

### Official Review · Reviewer_fWxe · 2024-12-03

**Novelty:** 4
**Technical Quality:** 4

**Review:**

This paper presents UniGO, a novel framework for modeling opinion evolution on graphs, tackling a complex and impactful problem within opinion dynamics and predictive modeling. The proposed use of a unified graph-based structure and a coarsen-refine mechanism demonstrates an innovative approach to addressing issues such as over-smoothing and capturing equilibrium states. While the paper is well-written and provides promising results, there are several areas where the clarity, contributions, and experimental rigor could be improved.

Wordy Introduction: The introduction repeats certain information, making it less concise. Refining this section to eliminate redundancy would improve readability and focus the narrative on the key motivations and contributions.

Vague Contributions: The contributions outlined in the introduction are not specific or compelling. For example, applying the framework to both synthetic and real-world datasets is not a unique contribution. The authors should emphasize the novelty of their methodology, insights, or theoretical contributions over the datasets used.

Missing Ablation Study: Although the method has multiple components, the paper does not include an ablation study to evaluate the individual contributions of these components (e.g., coarsen-refine mechanism, synthetic pretraining). Such an analysis is critical to demonstrate the value added by each part of the proposed framework.

Figure and Table Layout: The figures and tables consume excessive space. Resizing them appropriately would create room for additional analysis or visualizations, which could strengthen the results section.

**Questions:**

1. Can you clarify the specific novelty of UniGO compared to prior opinion dynamics models?
2. How does the coarsen-refine mechanism differ from existing graph neural network strategies for mitigating over-smoothing?
3. Why was an ablation study not included, and can you provide insights into the contribution of individual components in UniGO?
4. How were the synthetic datasets designed, and how do they reflect the diversity of real-world opinion dynamics?
5. What are the practical implications of this work for large-scale applications, and who are the intended users or stakeholders?

**Reviewer Confidence:**

4: The reviewer is certain that the evaluation is correct and very familiar with the relevant literature

**Scope:**

3: The work is somewhat relevant to the Web and to the track, and is of narrow interest to a sub-community

---

### Official Review · Reviewer_Au6Y · 2024-12-03

**Novelty:** 6
**Technical Quality:** 4

**Review:**

This submission presents a novel graph neural network framework for modeling opinion dynamics on networks. The work is original, however the clarity could be improved. For example, a " formal consistency" between opinion models dynamics and graph neural networks updates is claimed based on a vague similarity of the equations, but not formally proved. The fact that there isn't a formal equivalence is not necessarily a problem, but this should be clarified. Finally, the significance is not clear to me because there are not enough details about how the opinions from the empirical datasets are inferred. Hence, I cannot judge if the results are good or not.

pros:
- novel GNN framework
- method is well explained
- extensive training and testing
- apparent good results

cons:
- empirical tests are not detailed enough to judge the significance of the results (also, citations are missing for several datasets).
- formal equivalence between opinion models and GNN not clear.

**Questions:**

- Could you clarify how you constructed the opinion dynamic from the datasets?
- You say that you used "a combination of manual annotation and large language model", could you please give specifics details and report a measure of accuracy of the LLM annotations.
- Please give the references or details about collection methods for all the datasets.
- How did you filtered bot accounts?
- How did you choose the criteria for threshold of active users?
- How long is a "time step" in these datasets?
- How often are users' opinions updated?
- How much do opinions change over time?

**Reviewer Confidence:**

3: The reviewer is confident but not certain that the evaluation is correct

**Scope:**

4: The work is relevant to the Web and to the track, and is of broad interest to the community

---

### Official Review · Reviewer_qtwL · 2024-12-04

**Novelty:** 5
**Technical Quality:** 5

**Review:**

This paper presents UniGO, a unified framework that combines opinion dynamics theory with graph neural networks to model and predict opinion evolution on social networks. The work addresses several key challenges in opinion dynamics modeling, including the complexity of social interactions, diversity of opinion fusion rules, and difficulty in capturing equilibrium states while avoiding over-smoothing. The methodology is well-designed and thoroughly evaluated through both synthetic and real-world datasets. The work demonstrates strong originality in bridging the gap between theoretical opinion dynamics models and data-driven approaches, showing significant potential for practical applications in social network analysis.

Strengths:
1.	The paper proposes a unified opinion dynamics framework that effectively captures various opinion fusion rules, going beyond traditional single-rule models through a comprehensive integration of different theoretical models.
2.	The coarsen-refine mechanism successfully balances between preserving equilibrium phenomena and avoiding over-smoothing in GNNs, providing an innovative solution to a key technical challenge.
3.	The pretrain-finetune paradigm demonstrates strong practical value, showing good generalization capability from synthetic to real-world datasets and providing a viable solution for large-scale applications.

Weaknesses:
1.	The unified opinion dynamics model cannot handle coupled dynamics or temporal changes in graph structure, limiting its application in more complex scenarios.
2.	The real-world datasets used for validation are limited in scale and diversity, and the accuracy of transforming text data into opinion data remains a challenge.
3.	The model shows suboptimal performance on smaller datasets like the Delhi Election dataset, suggesting potential issues with model adaptation to different data scales.

**Questions:**

1.	About Model Scalability:
The experiments show that UniGO performs relatively worse on the Delhi Election dataset which has a smaller size. Could the authors provide more insights into how the model's performance scales with dataset size? What modifications could be made to improve performance on smaller datasets?
2.	Regarding Opinion Data Generation:
The paper notes that all real datasets are based on text data, making accurate transformation of text into opinion data challenging. Could the authors explain their current approach for converting text to opinion data and discuss potential improvements to this process? How does the quality of this transformation affect the model's performance?
3.	About the Coarsen-refine Mechanism:
While the coarsen-refine mechanism helps avoid over-smoothing, could the authors provide more theoretical analysis on how this mechanism preserves the equilibrium phenomena of opinion dynamics? What is the optimal ratio of supernodes to original nodes for different types of networks?
4.	Regarding Model Generalization:
The paper shows that pretraining on synthetic data helps with real-world performance. Could the authors discuss how sensitive the model is to the choice of synthetic data generation parameters? What guidelines would they suggest for generating synthetic data that best transfers to specific real-world scenarios?

**Reviewer Confidence:**

3: The reviewer is confident but not certain that the evaluation is correct

**Scope:**

4: The work is relevant to the Web and to the track, and is of broad interest to the community